# Heuristic Attention Representation Learning for Self-Supervised Pretraining

**DOI:** 10.3390/s22145169

**Published:** 2022-07-10

**Authors:** Van Nhiem Tran, Shen-Hsuan Liu, Yung-Hui Li, Jia-Ching Wang

**Affiliations:** 1Department of Computer Science and Information Engineering, National Central University, Taoyuan 3200, Taiwan; tvnhiemhmus@g.ncu.edu.tw (V.N.T.); 109522071@cc.ncu.edu.tw (S.-H.L.); jcw@csie.ncu.edu.tw (J.-C.W.); 2AI Research Center, Hon Hai Research Institute, Taipei 114699, Taiwan

**Keywords:** heuristic attention, perceptual grouping, self-supervised learning, visual representation learning, deep learning, computer vision

## Abstract

Recently, self-supervised learning methods have been shown to be very powerful and efficient for yielding robust representation learning by maximizing the similarity across different augmented views in embedding vector space. However, the main challenge is generating different views with random cropping; the semantic feature might exist differently across different views leading to inappropriately maximizing similarity objective. We tackle this problem by introducing **H**euristic **A**ttention **R**epresentation **L**earning (HARL). This self-supervised framework relies on the joint embedding architecture in which the two neural networks are trained to produce similar embedding for different augmented views of the same image. HARL framework adopts prior visual object-level attention by generating a heuristic mask proposal for each training image and maximizes the abstract object-level embedding on vector space instead of whole image representation from previous works. As a result, HARL extracts the quality semantic representation from each training sample and outperforms **existing** self-supervised baselines on several downstream tasks. In addition, we provide efficient techniques based on conventional computer vision and deep learning methods for generating heuristic mask proposals on natural image datasets. Our HARL achieves +1.3% advancement in the ImageNet semi-supervised learning benchmark and +0.9% improvement in AP_50_ of the COCO object detection task over the previous state-of-the-art method BYOL. Our code implementation is available for both TensorFlow and PyTorch frameworks.

## 1. Introduction

Visual representation learning has been an extended research area on supervised and unsupervised methods. Most supervised learning models learn visual representations by training with many labeled datasets, then transferring the knowledge to other tasks [1,2,3,4,5]. Most supervised learning frameworks try to tune their parameters such that they maximally compress mapping the particular input variables that preserve the information on the output variables [6,7,8]. As a result, most deep neural networks fail to generalize and maintain robustness if the test samples are different from the training samples on variant distribution and domains.

The new approaches are self-supervised representation learning to overcome the existing drawbacks of supervised learning [9,10,11,12,13,14,15]. These techniques have attracted significant attention for efficient, generalization, and robustness representation learning when transferring learned representation on multiple downstream tasks achieving on-par or even outperforming supervised baselines. Furthermore, self-supervised learning methods overcome the human supervision capability of leveraging the enormous availability of unlabeled data. Despite various self-supervised frameworks, these methods involve certain forms of the joint embedding architectures of the two branches neural network such as the Siamese network [16]. The neural networks of two branches are usually weights-sharing or different. In the joint embedding self-supervised framework, the common objective is to maximize the agreement between embedding vectors from different views of the same image. However, the biggest challenge is avoiding collapsing to a trivial constant solution, which is that all output embedding vectors are the same. Several strategies to prevent the collapsing phenomenon can be categorized into two main approaches: contrastive learning and non-contrastive learning. Self-supervised contrastive learning [9,17] prevents collapse via negative sample pairs. However, contrastive learning requires a large number of negative samples leading to the requirement of high computational resources. The efficient alternative approach is non-contrastive learning [13,14,18]. These frameworks rely only on positive pairs with a momentum encoder [13] or using an extra neural network on one branch with the block gradient flow [14,18]. 

Most existing contrastive and non-contrastive objectives are optimized based on the whole image semantic features across different augmented views. However, under this assumption, several challenges exist. First, popular contrastive methods such as SimCLR [9] and MoCo [17] require more computation and training samples than supervised methods. Second, more importantly, there is no guarantee that semantic representation of different objects will differentiate between different cropping views of the same image. For instance, several meaningful objects (vehicles, humans, animals, etc.) may exist in the same image. The semantic representation of vehicles and humans is different, so contrasting the similarity between different views based on the whole-image semantic feature may be misleading. Research in cognitive psychology and neural science [19,20,21,22] showed that early visual attention helps humans focus on the main group of important objects. In computer vision, the perceptual grouping principle is used to group visual features into meaningful parts that allow a much more effective learning representation of the input context information [21].

Motivated by perceptual grouping, we proposed the **H**euristic **A**ttention **R**epresentation **L**earning (HARL) framework that comprises two main components. First, the early attention mechanism uses unsupervised techniques to generate the heuristic mask to extract object-level semantic features. Second, we construct a framework to abstract and maximize similarity object-level agreement (foreground and background) across different views beyond augmentations of the same image [13,18,23]. This approach helps enrich the quantity and quality of semantic representation by leveraging foreground and background features extracted from the training dataset.

We can summarize our main findings and contributions as follows:We introduce a new self-supervised learning framework (HARL) that maximizes the similarity agreement of object-level latent embedding on vector space across different augmented views. The framework implementation is available in the Appendix A section.We utilized two heuristic mask proposal techniques from conventional computer vision and unsupervised deep learning methods to generate a binary mask for the natural image dataset.We construct the two novel heuristic binary segmentation mask datasets for the ImageNet ILSVRC-2012 [24] to facilitate the research in the perceptual grouping for self-supervised visual representation learning. The datasets are available to download in the Data Availability Statement section.Finally, we demonstrate that adopting early visual attention provides a diverse set of high-quality semantic features that increase more effective learning representation for self-supervised pretraining. We report promising results when transferring HARL’s learned representation on a wide range of downstream vision tasks.

The remainder of this paper is organized as follows. In Section 2, we discussed related works. Section 3 introduces the HARL framework in detail. Section 4.1 briefly describes the implementation of the HARL framework in self-supervised pretraining. Section 4.2 evaluates and benchmarks HARL performance on the ImageNet evaluation, transfers learning to other downstream tasks and compares it to previous state-of-the-art methods. In Section 5, we provide the analysis of the components impacting the performance and understanding of the behavior of our proposed method. Finally, this paper is concluded in Section 6.

## 2. Related Works

Our method is mostly related to unsupervised visual representation learning methods, aiming to exploit input signals’ internal distributions and semantic information without human supervision. The early works focused on several design-solving pretext tasks, and image generation approaches. Pretext tasks focus on the aspects of image restoration such as denoising [25], predicting noise [26], colorization [27,28], inpainting [29], predicting image rotation [30], solving jigsaw puzzles [31] and more [32,33]. However, these methods, the learned representation of neural networks pre-trained on pretext tasks, still failed in generalization and robustness when performed on different downstream tasks. The generative adversarial learning [34,35,36] and variational auto-encoding [25,37,38] operate directly on pixel space and high-level details for image generations, which require costly computation that may not be essential and efficient for visual representation learning.

**Self-supervised contrastive learning.** The popular self-supervised contrastive learning frameworks [9,39,40] aim to pull semantic features from different cropping views of the same image while pushing other features away from other images. However, the downside of contrastive methods is that they require a considerable number of negative pairs, leading to significant computation resources and memory footprint. The efficient alternative approach is non-contrastive learning [13,18], which only maximizes the similarity of two views from the same image without contrast to other views from different images.

**Self-supervised non-contrastive learning.** Distillation learning-based framework [13,18] inspired by knowledge distillation [41] is applied to joint embedding architecture. One branch is defined as a student network, and another is described as a teacher network. The student network is trained to predict the representation of the teacher network; the teacher network’s weights are optimized from the student network by a running average of the student network’s weights [13] or by sharing with the student’s weights and blocking the gradient flow through the teacher network [18]. Non-contrastive frameworks are effective and computationally efficient compared to self-supervised contrastive frameworks [9,17,39].

However, most contrastive or non-contrastive self-supervised techniques maximize similarity agreements of the whole-image context representation of different augmented views. While developing localization attention to separate the semantically features [42,43] by the perceptual grouping of semantic information proved that adopting prior mid-level visible in pretraining gains efficiency for representation learning. The most recent study related to our [39] leveraging visual attention with segmentation obtained impressive results when transferring the learned representation to downstream tasks on object detection and segmentation in multiple datasets. In contrast to our work, previous work employs pixel-level models for contrastive learning, which uses backbones specialized for semantic segmentation and uses different loss functions. It is important to note that the primary work objective is difficult to transfer to other self-supervised frameworks. It also did not investigate the masking feature method or the impact of the dimension and size of the output spatial feature maps on the latent embedding representation, which we will examine next.

## 3. Methods

In contrastive or non-contrastive learning-based frameworks, HARL object-level objectives are applicable. For example, our study implements a non-contrastive learning framework using an exponential moving average weight parameter of one encoder to another and an extra predictor inspired by BYOL [13]. HARL’s objective maximizes the agreement of the object-level (foreground and background) latent embedding vector across different cropping views beyond augmentations shown in Figure 1.

### 3.1. HARL Framework

The HARL framework consists of three essential steps. In step 1, we estimate the heuristic binary mask for the input image, which segments an image into foreground and background (see described detail in Section 3.2). Next, these masks can be computed using either conventional computer vision methods such as DRFI [44] or unsupervised deep learning saliency prediction [42]. After the mask is estimated, we perform the same image transformation (cropping, flipping, resizing, etc.) to both the image and its mask. Finally, if it is the RGB image, transformations such as color distortion can be applied to the image, such as the image augmentation pipeline of SimCLR [9]. The detailed augmentation pipeline is described in Section A.1. After data augmentation, each image and mask pair generated two augmented images x, x′ aligned with two augmented masks m and m′ as illustrated in Figure 1.

In step 2, we implement standard ResNet-50 [45] convolution residual neural network for feature extractor denotation as ƒ. Each image through the feature extractor encodes the output to obtain the spatial feature maps of size 7 × 7 × 2048, and this feature extraction process can be formulated as *h* = ƒ(x), where *h* ∈ ℝH×W×D.. Then, the feature maps can be separated into the foreground and background feature maps by performing element-wise multiplication with the heuristic binary mask. In addition, we provide ablation studies to analyze the impact of the spatial feature map in various sizes and dimensions, as described in Section 5.1. The foreground and background features are denoted as, hf
hb (Section A.2 provides detail of the masking feature method). The foreground and background spatial features are down-sampled using global average pooling to project to a smaller dimension with non-linear multi-layer perceptron (MLP) architecture g.

HARL framework structure adapts from BYOL [13], in which one augmented image (x) is processed with the encoder fθ, and projection network gθ, where θ is the learned parameters. Another augmented image (x′) is processed with fξ and gξ, where ξ is an exponential moving average of θ. The first augmented image is further processed with the predictor network qθ. The projection and predictor network architectures are the same using the non-linear multi-layer perceptron (MLP), as detailed in Section 4. The definition of encoder, projection, and prediction network is adapted from the BYOL. Finally, the latent representation embedding vectors corresponding to the augmented image’s foreground and background features are denoted as zf, zb,zf′ and zb′∈ℝd.
where:                                            zf, zb≜gθ°qθ(hf,hb),                                                     zf′, zb′≜gξ(hf′,hb′).

In step 3, we compute the HARL’s loss function of the given foreground and background latent representations ($z_{f},\;z_{b},z_{f^{\prime }}$ and $z_{b^{\prime }}$ are extracted from two augmented images x, x′) which is defined as mask loss, as illustrated in Equation (1). We apply $\ell_{2}$-normalization to these latent vectors, then minimize their negative cosine similarity agreement with the weighting coefficient α . We study the impact of α value and the combination of the whole image and object-level latent embedding vector in the loss objective provided in Section 5.2.
(1)$${ℒ}_{\theta }^{Maskloss}=-\left(\alpha \cdot \frac{z_{f}}{\|z_{f}{\|}_{2}}\cdot \frac{z_{f^{\prime }}}{\|z_{f^{\prime }}{\|}_{2}}+\left(1-\alpha \right)\cdot \frac{z_{b}}{\|z_{b}{\|}_{2}}\cdot \frac{z_{b^{\prime }}}{\|z_{b^{\prime }}{\|}_{2}}\right),$$
where ${\left\|.\right\|}_{2}$ is $\ell_{2}$-norm, and it is equivalent to the mean squared error of $\ell_{2}$-normalized vectors. The weighting coefficient α is in the range [0–1].

We symmetrized loss ℒ by separately feeding augmented image and mask of view one to the online network and augmented image and mask of view two to the target network and vice versa to compute the loss at each training step. We perform a stochastic optimization step to minimize the symmetrized loss ${ℒ}_{symmetrized}$ = ℒ + ${ℒ}^{\sim }$.
(2)$${ℒ}_{symmetrized}={ℒ}_{\theta }^{Maskloss}+{ℒ}_{\theta }^{\sim Maskloss}.$$

After pretraining processing is complete, we only keep the encoder ${}_{\theta }$ and discard all other parts of the networks. The whole training procedure summary is in the python pseudo-code Algorithm 1.
**Algorithm 1: HARL: Heuristic Attention Representation Learning**Input:
   
   
D
, M
, T$,\;\mathrm{and}\;T^{\prime }$: set of images, mask and distributions of transformations
   
   
θ$,\;f_{\theta }$$,\;g_{\theta }$$,\;\mathrm{and}\;Q_{\theta }\;:$ initial online parameters, encoder, projector, and predictor   
    ξ$,\;f_{\xi }$$,\;g_{\xi }$; // initial target parameters, target encoder, and target projector   
    Optimizer; //optimizer, updates online parameters using the loss gradient   
    K and N; //total number of optimization steps and batch size   
   ${\left\{T_{K}\right\}}_{k=1}^{K}$$\;\mathrm{and}\;{\left\{\eta_{k}\right\}}_{k=1}^{K}$; //target network update schedule and learning rate  schedule
 1:  For k = 1 to K do 2:   $B\leftarrow {\left\{x_{i}∼D\right\}}_{i=1}^{N};$ //sample a batch of N images 3:   $C\leftarrow {\left\{m_{i}∼M\right\}}_{i=1}^{N};$ //sample a batch of N mask 4:    For $x_{i}\in B,m_{i}\in C\;\;$ 5:    $h\leftarrow f_{\theta }\left(t\left(x_{i}\right)\right);$ //compute the encoder feature map 6:    $h\prime \leftarrow f_{\theta }\left(t^{\prime }\left(x_{i}\right)\right);$ //compute the target encoder feature map 7:    $h_{f},h_{b}\leftarrow m_{i}*h;$ //separate the feature map 8:    $h_{f^{\prime }},h_{b\prime }\leftarrow m_{i}*h^{\prime };$ //separate the target feature map 9:    $z_{f},z_{b}\leftarrow q_{\theta }\left(g_{\theta }\left(h_{f},h_{b}\right)\right);$ //compute projections 10:    $z_{f\prime },z_{b\prime }\leftarrow g_{\xi }\left(h_{f\prime },h_{b\prime }\right);$ compute target projections 11:    $l_{i}\leftarrow -2\cdot \left(\alpha \cdot \frac{z_{f}}{\|z_{f}\|{}_{2}}\cdot \frac{z_{f}}{\|z_{f}\|{}_{2}}+\left(1-\alpha \right)\cdot \frac{z_{b^{\prime }}}{\|z_{b^{\prime }}\|{}_{2}}\cdot \frac{z_{b^{\prime }}}{\|z_{b^{\prime }}\|{}_{2}}\right);$//compute loss 12:  End for 13:  $\delta \theta \leftarrow \frac{1}{N}\Sigma_{N}^{i=1}\;\partial l_{i}$
 //compute the total loss gradient w.r.t. θ 14:  $\theta \leftarrow optimizer\left(\theta ,\delta \theta ,\eta_{k}\right);$ //update online parameters 15:  $\xi \leftarrow \tau_{k}\xi +\left(1-\tau_{k}\right)\theta ;$ //update target parameters encoder $f_{\theta }$



### 3.2. Heuristic Binary Mask

Our heuristic binary mask estimation technique does not rely on external supervision, nor is it trained with the limited annotated dataset. We proposed two approaches using conventional computer vision and unsupervised deep learning to carry it out, and these methods appear to be well generalized for various image datasets. First, we use the traditional computer vision method DRFI [44] to generate a diverse set of binary masks by varying the two hyperparameters (the Gaussian filter variance σ and the minimum cluster size s). In our implementation, we defined σ = 0.8 and s = 1000 for generating binary masks in the ImageNet [24] dataset. In the second approach, we leverage the self-supervised encoder feature extractor of the pre-trained ResNet-50 backbone from [9,42], then pass the output feature maps into a 1 × 1 convolutional classification layer for saliency prediction. The classification layer predicts the saliency or “foregroundness” of a pixel. Therefore, we take the output values of the classification layer and set a threshold of 0.5 to decide which pixels belong to the foreground. Pixel saliency values greater than the threshold are determined as foreground objects. Figure 2 shows the example heuristic mask estimated by these two methods. The detailed implementation of the two methods, DRFI and deep learning feature extractor combined with 1 × 1 convolutional layer is described in Appendix D. In most of our experiments, we used the mask generated by the deep learning method because it is faster than DRFI by running with GPU instead of only with CPU.

## 4. Experiments

### 4.1. Self-Supervised Pretraining Implementation

HARL is trained on RGB images and the corresponding heuristic mask of the ImageNet ILSVRC-2012 [24] training set without labels. We implement standard encoder ResNet [45]. According to previous works by SimCLR and BYOL [9,13], the encoder representation output is projected into a smaller dimension using a multi-layer perceptron (MLP). In our implementation, the MLP comprised a linear layer with an output size of 4960 followed by batch normalization [46], rectified linear units (ReLU) [47] and the final linear layer with 512 output units. We apply the LARS optimizer [48] with the cosine decay learning rate schedule without restarts [49], over 1000 epochs on the base learning rate of 0.2, scaled linearly [50] with the batch size (LearningRate = 0.2 × BatchSize/256) and the warmup epochs of 10. Furthermore, we apply a global weight decay parameter of 5 × 10^−7^ while excluding the biases and normalization parameters from the LARS adaptation and weight decay. The optimization of the online network and target network follow the protocol of BYOL [13]. We use a batch size of 4096 splits over 8 Nvidia A100GPUs. This setup takes approximately 149 h to train a ResNet-50 (×1).

The computational self-supervised pretraining stage requirements are largely due to forward and backward passes through the convolutional backbone. For the typical ResNet-50 architecture applied to 224 × 224 resolution images, a single forward pass requires approximately 4B FLOPS. The projection head MLP (2048 × 4096 + 4096 × 512) requires roughly 10M FLOPS. In our implementation, the convolution network backbone and MLP network are similar compared to baselines BYOL. Since we forward to the foreground and background representation through the projection head two times instead of one, it results in an additional 10M FLOPS in our framework, less than 0.25% of the total. Finally, the cost of computing the heuristic mask images is negligible because they can be computed once and reused throughout training. Therefore, the complexity of each iteration between our method and the baseline BYOL is almost the same for “computational cost” and “training time”.

### 4.2. Evaluation Protocol

We evaluate the learned representation from the self-supervised pretraining stage on various natural image datasets and tasks, including image classification, segmentation and object detection. First, we assess the obtained representation on the linear classification and semi-supervised learning on the ImageNet following the protocols of [9,51]. Second, we evaluate the generalization and robustness of the learned representation by conducting transfer learning to other natural image datasets and other vision tasks across image classification, object detection and segmentation. Finally, in Appendix C, we provide a detailed configuration and hyperparameters setting of the linear and fine-tuning protocol in our transfer learning implementation.

#### 4.2.1. Linear Evaluation and Semi-Supervised Learning on the ImageNet Dataset

The evaluation for linear and semi-supervised learning follows the procedure in [9,52,53]. For the linear evaluation, we train a linear classifier on top of the frozen encoder representation and report Top-1 and Top-5 accuracies in percentage for the test set, as shown in Table 1. We then evaluate semi-supervised learning, which is fine-tuning the pre-trained encoder on a small subset with 1% and 10% of the labeled ILSVRC-2012 ImageNet [24] training set. We also report the Top-1 and Top-5 accuracies for the test set in Table 1. HARL obtains 54.5% and 69.5% in Top-1 accuracy for semi-supervised learning using the standard ResNet-50 (×1). It represents a +1.3% and +0.7% advancement over the baseline framework BYOL [13] and significant improvement compared to the strong supervised baseline in the accuracy metric.

#### 4.2.2. Transfer Learning to Other Downstream Tasks

We evaluated the HARL’s quality of representation learning on linear classification and fine-tuned model following the evaluation setup protocol [9,13,39,55] as detailed in Section B.2. HARL’s learned representation can perform well for all six different natural distribution image datasets. It has competitive performance in various distribution datasets compared to baseline BYOL [13] and improves significantly compared to the SimCLR [9] approach over six datasets, as shown in Table 2.

We further evaluated HARL’s generalization ability and robustness with different computer vision tasks, including object detection of VOC07 + 12 [56] using Faster R-CNN [57] architecture with R50-C4 backbone and instance segmentation task of COCO [58] using Mask R-CNN [59] with R50-FPN backbone. The fine-tuning setup procedure and setting hyperparameter are detailed in Section B.3. We report the performance of the standard AP, AP_50_ and AP_75_ metrics in Table 3. HARL outperforms the baselines BYOL [13] and also has a significantly better performance than other self-supervised frameworks such as SimCLR [9], MoCo_v2 [17] and supervised baseline on object detection and segmentation.

## 5. Ablation and Analysis

We study the HARL’s components to give the intuition of its behavior and impact on performance. We reproduce the HARL framework with multiple running experiments. For this reason, we hold the same set of hyperparameter configurations and change the configuration of the corresponding component, which we try to investigate. We perform our ablation experiments on the ResNet-50 and ResNet-18 architecture on the ImageNet training set without labels. We evaluate the learned representation on the ImageNet linear evaluation during the self-supervised pretraining stage. To do so, we attach the linear classifier on top of the base encoder with the block gradient flow on the linear classifier’s input, which stops influencing and updating the encoder with the label information (a similar approach to SimCLR [9]). We run ablations over 100 epochs and evaluate the performance of the public validation set of the original ILSVRC2012 ImageNet [24] in the Top-1 accuracy metric at every 100 or 200 steps per epoch following the protocol as described in Section B.1.

### 5.1. The Output of Spatial Feature Map (Size and Dimension)

In our HARL framework, separating foreground and background features from the output spatial feature map is essential to maximize the similarity objective across different augmented views. To verify this hypothesis, we analyze several spatial outputs in various sizes and dimensions by modifying the ResNet kernel’s stride to generate the different feature map sizes with the same dimension. For illustration, the standard ResNet is the sequence of four convolution building blocks (conv2_x, conv3_x, conv4_x, conv5_x). For ResNet-50 architecture, the dimension of conv_5x block output feature map is 7 × 7 × 2048. After changing the kernel stride of the conv_4x block from two to one, its new dimension will be 14 × 14 × 2048. In this modified ResNet-50 architecture, the conv5_x block’s spatial feature map size is the same as the conv4_x block output.

We conduct the experiment for three different sizes including a deep ResNet-50 (7 × 7 × 2048, 14 × 14 × 2048, 28 × 28 × 2048) and a shallow ResNet-18 (7 × 7 × 512, 14 × 14 × 512, 28 × 28 × 512). Figure 3 shows the experimental results of various output sizes and dimensions in the pretraining stage that impact the learned representation when evaluating transfer representation on the ImageNet with linear evaluation protocol. Both shallow and deep ResNet architecture yields better learning ability on the larger output spatial feature map size 14 × 14 than 7 × 7. In our experiments, the performance decreases as we continue to go to a larger output size, 28 × 28 or 56 × 56.

### 5.2. Objective Loss Functions

HARL framework structure reuses elements of BYOL [13]. We use two neural networks denoted as *online network* and *target network*. Each network is defined by a set of parameters θ and ξ. The optimization objective minimizes the loss ${ℒ}_{\theta ,\;\xi }$ with respect to learnable parameters θ, while the set ξ is parameterized by using an exponential moving average of the θ, as shown in Equation (3):(3)ξ← τξ+(1−τ)θ.

Unlike previous approaches that minimize loss function only based on the whole image latent embedding vector between two augmented views, HARL minimized the similarity of object-level latent representation, which associated the same spatial regions abstracting from segmentation mask and thus same semantic meanings. As shown in Figure 1, we use the mask information to separate the spatial semantic object-level feature (foreground and background) of the two augmented views. Then, we minimize their negative cosine similarity, denoting mask loss in Equation (1). In addition to our mask loss objective, we combine the distance loss of the whole image representation and object-level, resulting in hybrid loss as described in Equation (4). We study these two loss objectives in the self-supervised pretraining stage and then evaluate the obtained representation on the ImageNet with a linear evaluation protocol.

#### 5.2.1. Mask Loss

The mask loss objective converges to minimizing the distance loss objective between foreground and background latent embedding on vector space ${ℒ}_{\mathrm{foreground}\left(\theta ,\;\xi \right)}$ and ${ℒ}_{\mathrm{background}\left(\theta ,\;\xi \right)}$ with the weighting coefficient α as described in Equation (1). We study the impact of α when it is set to a few predefined values and when it varies according to the cosine scheduling rule. In the first approach, we perform self-supervised pretraining sweeping over three different values {0.3, 0.5, 0.7}. In the second approach, we schedule the α based on a cosine schedule, $\alpha \;≜\left(1-\left(1-\alpha_{base}\right)\right)\cdot (\cos\pi k/K)+1)/2,$ to gradually increase from the starting $\alpha_{base}$ value to 1 corresponding current training step k over total training step K. We tried three $\alpha_{base}$ values, including 0.3, 0.5 and 0.7. We report the Top-1 accuracy on the ImageNet linear evaluation set during the self-supervised pretraining stage, as shown in Figure 4. The weighting coefficient α value of 0.7 yields the consistent learned representation of both approaches. Furthermore, the experimental results demonstrate that the foreground is more important than the background latent representation. For example, in the ImageNet training set, many images exist in which the background information is more than 50% of the image.

#### 5.2.2. Hybrid Loss

The objective combines whole image representation embedding v1 and v2 together with object-level representation embedding mask loss described in Equation (1). v1 and  v2 are extracted from the two augmented views x and x’ and are denoted as v1≜θ °gθ°qθ(x)  ℝd and v2≜ ξ °gξ( x′)  ℝd. The hybrid loss minizines the negative cosine similarity with weighting coefficient λ:(4)ℒθhybrid=−[λ·v1‖v1‖2·v2‖v2‖2+(1−λ)·ℒθMaskloss],
where v1 and  v2 are the whole image latent representation; ℒθMaskloss is the distance loss computed from the foreground and background latent representation described in Equation (1); ‖.‖2 is ℓ2-norm; and λ is the weighting coefficient in the range [0–1].

To study the impact of weighting coefficient λ, we use a cosine scheduling value similar to α in the mask loss section. In our experiment, the weighting coefficient λ cosine scheduling sweeping over four $\lambda_{base}$ values {0.3, 0.6, 0.7, 0.9}. We report the Top-1 accuracy of the ImageNet linear evaluation protocol on the validation set during the self-supervised pretraining stage, shown in Figure 5. We found using the weighting coefficient $\lambda_{base}$ value of 0.7 obtains the consistent learned representation when transferring to downstream tasks.

#### 5.2.3. Mask Loss versus Hybrid Loss

We compare the obtained representation using mask loss and hybrid loss on self-supervised pretraining. To do so, we implement the HARL framework with both loss objectives on self-supervised pretraining. We use the cosine schedule function to control the weighting coefficient α and λ sweeping on three different initial values {0.3, 0.5, 0.7} for both coefficients. We evaluate the obtained representation of the pre-trained encoder using ResNet-50 backbone in Top-1 and Top-5 accuracy (in%) on ImageNet linear evaluation protocol, as shown in Table 4. According to the experimental result, using the hybrid loss incorporated between global and object-level latent representation yields better representation learning during self-supervised pretraining.

### 5.3. The Impact of Heuristic Mask Quality

In our work, the HARL objective uses two different image segmentation techniques. Which ones lead to the best representation? We first consider the heuristics mask retrieving from the computer vision DRFI [44] approach by varying the two hyperparameters (the Gaussian filter variance σ and the minimum cluster size s) as described in detail in Section C.1. In our implementation, we generate a diverse set of binary masks by different combinations of σ ∈ {0.2, 0.4, 0.8} and c ∈ {1000, 1500}. The sets of the generated masks are shown in Figure 6. We found that the setting of σ = 0.8 and s = 1000 generate more stable mask quality than other combinations. Following the deep learning technique, we use the pre-trained deep convolution neural network as the feature extractor and design a saliency head prediction on top of the feature extractor output’s representation in the following three steps described in Section C.2. The generated masks are dependent on the pixel saliency threshold, which determines the foregroundness and backgrounness of the pixel. In our implementation, we tested the saliency threshold value ranging in {0.4, 0.6, 0.7} as shown in Figure 7. We choose the threshold value equal to 0.5 for generating masks in the ImageNet dataset. After choosing the best configure of the two techniques, we generate the mask for the whole training set of the ImageNet [24] dataset. We evaluate the mask quality generated by computing the mean Intersection-Over-Union (mIoU) between masks generated with the ImageNet ground-truth mask annotated by humans from Pixel-ImageNet [60]. The mIoU of the deep learning masks achieves 0.485 over 0.398 of DRFI masks on the subset of 0.485 million images (946/1000 classes of ImageNet). We found that in a complex scene, where multiple objects exist in a single image, the mask generated from the DRFI technique is noisier and less accurate than the deep learning masks, as illustrated in Figure 8.

To fully evaluate the impact of representation learning on downstream performance, we inspect the obtained representational quality with the transfer learning performance on the object detection and segmentation shown in Table 5. The result indicates that for most object detection and segmentation tasks, HARL learning based on masks with deep learning outperforms the one with DRFI masks, although the difference is very small. It shows that the quality of the mask used for HARL does have a small impact on the performance of the downstream task.

## 6. Conclusions and Future Work

We introduce the HARL framework, a new self-supervised visual representation learning framework, by leveraging visual attention with the heuristic binary mask. As a result, HARL manages higher-quality semantical information that considerably improves representation learning of self-supervised pretraining compared to previous state-of-the-art methods [9,13,17,18] on semi-supervised and transfers learning on various benchmarks. The two main advantages of the proposed method include: (i) the early attention mechanism that can be applied across different natural image datasets because we use unsupervised techniques to generate the heuristic mask and do not rely on external supervision; (ii) the entire framework can transfer and adapt quickly either to self-supervised contrastive or non-contrastive learning framework. Furthermore, our method will apply and accelerate the currently self-supervised learning direction on pixel-level objectives. Our object-level abstract will make this objective more efficient than the existing work based on computing pixel distance [61].

In our HARL framework, the heuristic binary mask is critical. However, the remaining challenge of estimating accurate masks is suitable for datasets with one primary object, such as the ImageNet dataset. The alternative is mining the object proposal of the image in the complex dataset which contains multiple things by producing heuristic semantic segmentation masks. Designing the new self-supervised framework to solve the remaining challenge of datasets which contain multiple objects is an essential next step and exciting research direction for our future work.

## Figures and Tables

**Figure 1 sensors-22-05169-f001:**
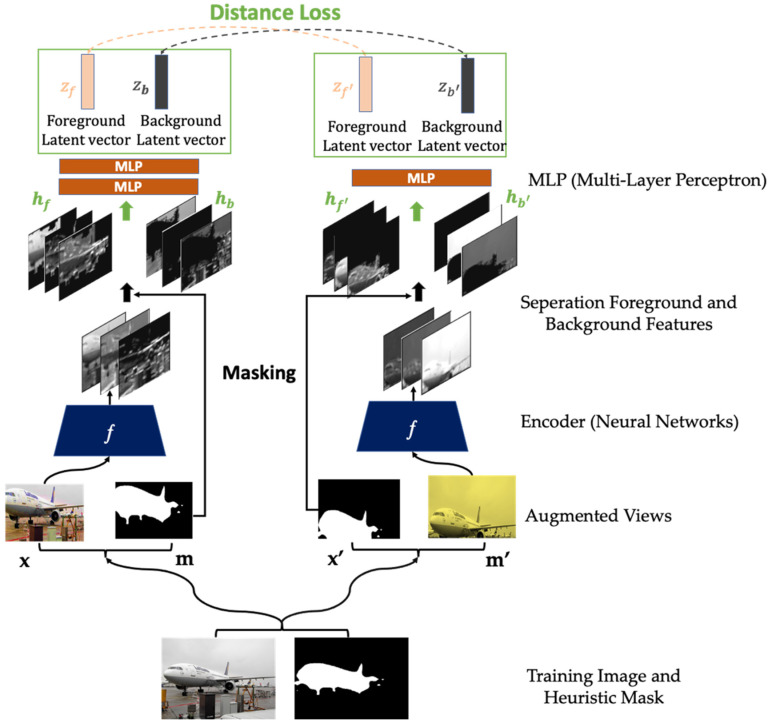
The HARL’s architecture. The heuristic binary mask can be estimated by using either conventional computer vision or deep learning approaches. After that, data augmentation transformation is applied to both the image and its mask (**bottom**). Then, the image pairs flow to a convolutional feature extraction module. The heuristic mask is used to mask the feature maps (which are the outputs of the feature extraction module) in order to separate the foreground from the background features (**middle**). These features are further processed by non-linear multi-layer perceptron modules (MLP). Finally, the similarity objective maximizes foreground and background embedding vectors across different augmented views from the same image (**top**).

**Figure 2 sensors-22-05169-f002:**
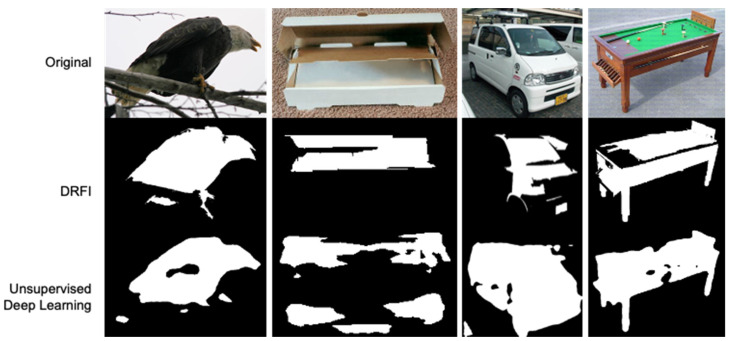
Example of heuristic binary masks used for mask contrastive learning framework. First row: random images from the ImageNet [24] training set. Second row: mask generated based on DRFI algorithm with a predefined sigma σ value of 0.8 and component size values of 1000. The third row is the mask obtained from the self-supervised pre-trained feature extractor ResNet-50 backbone directly followed by a 1 × 1 convolutional classification foreground and background prediction.

**Figure 3 sensors-22-05169-f003:**
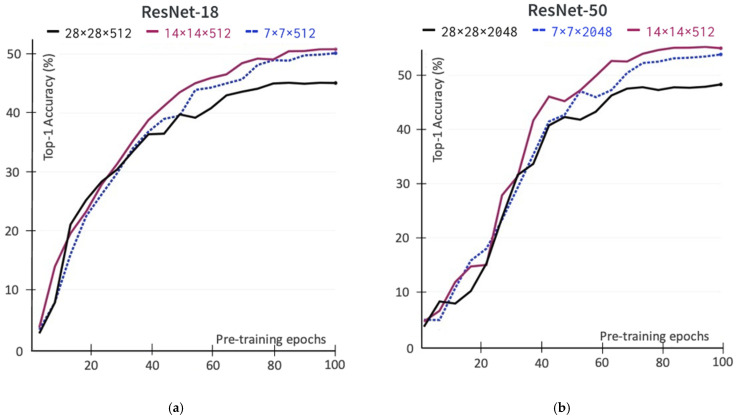
The ImageNet linear evaluation Top-1 accuracy (in %) of spatial output feature maps in various sizes and dimensions during the self-supervised pretraining stage. (**a**) The self-supervised pre-trained encoder uses the ResNet-18 backbone; (**b**) The self-supervised pre-trained encoder uses the ResNet-50 backbone.

**Figure 4 sensors-22-05169-f004:**
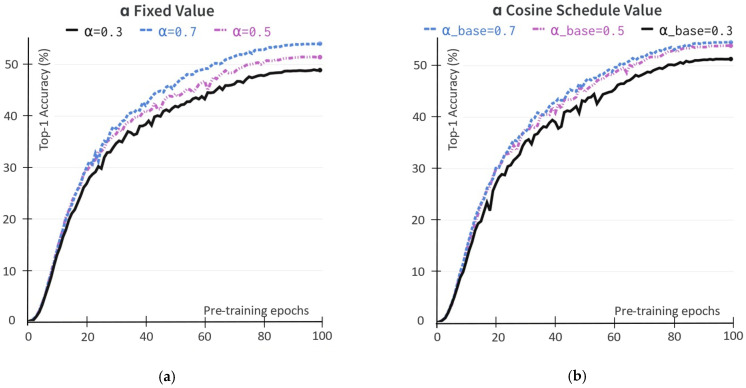
The impact of weighting coefficient α value to the obtained representation during the self-supervised pretraining stage with the ResNet-50 backbone. The evaluation during pretraining uses the ImageNet linear evaluation protocol in Top-1 accuracy (in%). (**a**) The α value is the fixed value; (**b**) The α value follows the cosine function scheduler.

**Figure 5 sensors-22-05169-f005:**
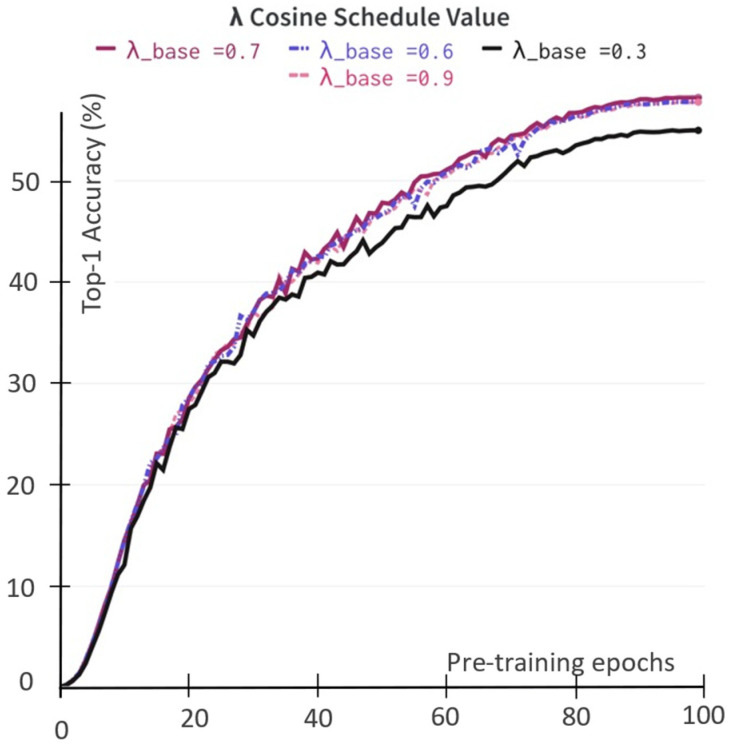
The impact of the weighting coefficient λ value to the obtained representation of the pre-trained encoder (ResNet-50) during the self-supervised pretraining stage on the ImageNet linear evaluation protocol in Top-1 accuracy (in%).

**Figure 6 sensors-22-05169-f006:**
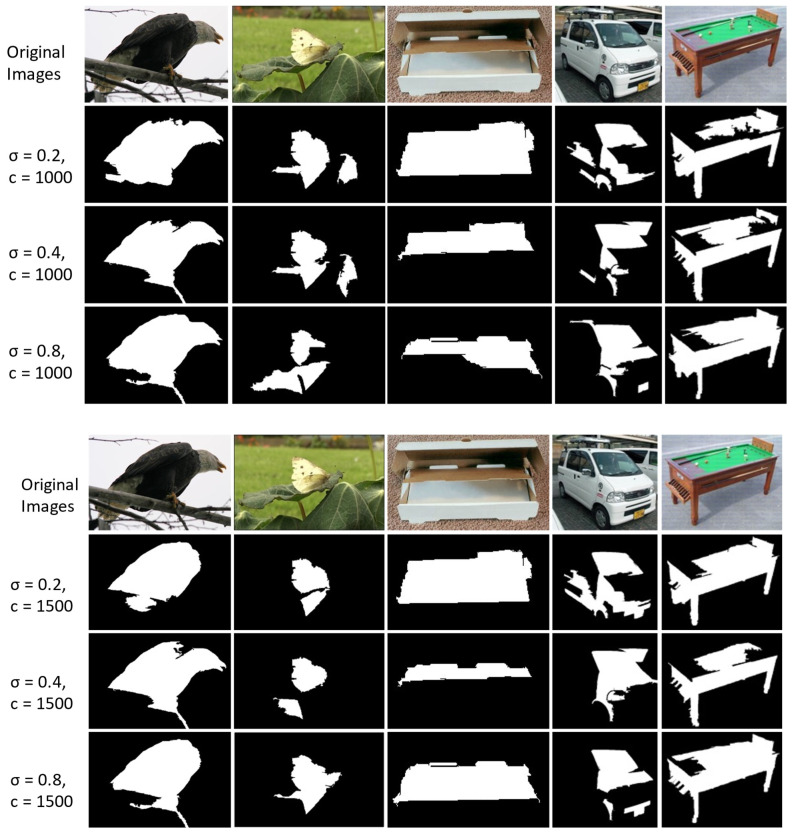
The heuristic binary masks are generated using DRFI with σ = {0.2, 0.4, 0.8} with c = {1000, 1500}.

**Figure 7 sensors-22-05169-f007:**
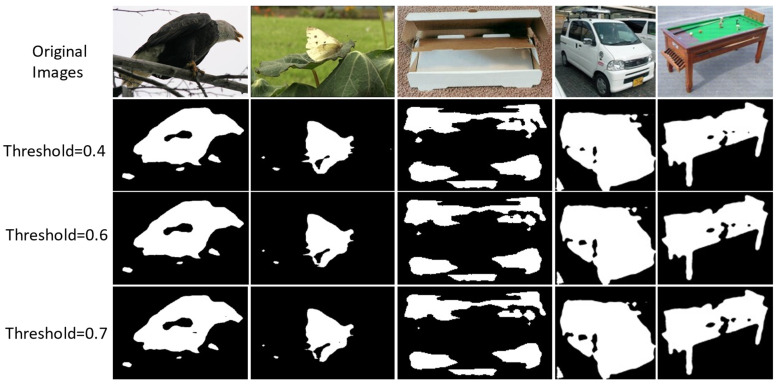
The heuristic binary masks are generated using an unsupervised deep learning encoder with saliency threshold values.

**Figure 8 sensors-22-05169-f008:**
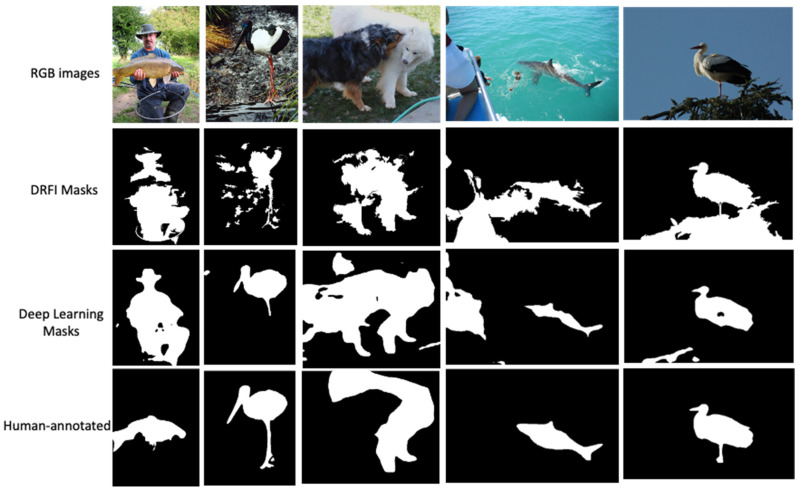
The inspection examples of the generated heuristic binary masks between DRFI and deep learning.

**Table 1 sensors-22-05169-t001:** **Evaluation on the ImageNet.** The linear evaluation and semi-supervised learning with a fraction (1% and 10%) on ImageNet labels report Top-1 and Top-5 accuracies (in%) using the pre-trained ResNet-50 backbone. The best result is bolded.

Method	Linear Evaluation			Semi-Supervised Learning		
	Top-1	Top-5		Top-1	Top-5	
			1%	10%	1%	10%
Supervised	76.5	-	25.4	56.4	48.4	80.4
PIRL [11]	63.6	-	-	-	57.2	83.8
SimCLR [9]	69.3	89.0	48.3	65.6	75.5	87.8
MoCo [17]	60.6	-	-	-	-	-
MoCo v2 [54]	71.1	-	-	-	-	-
SimSiam [18]	71.3	-	-	-	-	-
BYOL [13]	74.3	91.6	53.2	68.8	78.4	89.0
HARL (ours)	74.0	91.3	**54.5**	**69.5**	**79.2**	**89.3**

**Table 2 sensors-22-05169-t002:** **Transfer via fine-tuning on the image classification task.** The transfer learning performance between HARL framework and other self-supervised baseline benchmarks across six natural image classification datasets with the self-supervised pre-trained representation on the ImageNet 1000 classes using the standard ResNet-50 backbone. The best result is bolded.

Method	Food101	CIFAR10	CIFAR100	SUN397	Cars	DTD
Linear evaluation:						
HARL (ours)	75.0	**92.6**	77.6	61.4	67.3	**77.3**
BYOL [13]	**75.3**	91.3	**78.4**	**62.2**	**67.8**	75.5
MoCo v2 (repo)	69.2	91.4	73.7	58.6	47.3	71.1
SimCLR [9]	68.4	90.6	71.6	58.8	50.3	74.5
Fine-tuned:						
HARL (ours)	88.0	97.6	**85.6**	**64.1**	91.1	**78.0**
BYOL [13]	**88.5**	97.4	85.3	63.7	**91.6**	76.2
MoCo v2 (repo)	86.1	97.0	83.7	59.1	90.0	74.1
SimCLR [9]	88.2	**97.7**	85.9	63.5	91.3	73.2

**Table 3 sensors-22-05169-t003:** **Transfer learning to other downstream vision tasks.** Benchmark the transfer learning performance between HARL framework and other self-supervised baselines on object detection and instance segmentation task. We use Faster R-CNN with C4 backbone for object detection and Mask-RCNN with FPN backbone for instance segmentation. Object detection and instance segmentation backbone initialize with the pre-trained ResNet-50 backbone on ImageNet 1000 classes. The best result is bolded.

Method	Object Detection						Instance Segmentation		
	VOC07 + 12 Detection			COCO Detection			COCO Segmentation		
	AP_50_	AP	AP_75_	AP_50_	AP	AP_75_	${\mathrm{AP}}_{50}^{mask}$	${\mathrm{AP}}^{mask}$	${\mathrm{AP}}_{75}^{mask}$
Supervised	81.3	53.5	58.8	58.2	38.2	41.2	54.7	33.3	35.2
SimCLR-IN [18]	81.8	55.5	61.4	57.7	37.9	40.9	54.6	33.3	35.3
MoCo [17]	82.2	57.2	63.7	58.9	38.5	42.0	55.9	35.1	37.7
MoCo v2 [54]	82.5	57.4	64.0	-	39.8	-	-	36.1	-
SimSiam [18]	82.4	57.0	63.7	59.3	39.2	42.1	56.0	34.4	36.7
BYOL [13]	-	-	-		40.4	-	-	37.0	-
BYOL (repo)	82.6	55.5	61.9	61.2	40.2	43.9	58.2	36.7	39.5
HARL (ours)	**82.7**	56.3	62.4	**62.1**	**40.9**	**44.5**	**59.0**	**37.3**	**40.0**

**Table 4 sensors-22-05169-t004:** The comparison obtained representation of HARL framework using mask loss and hybrid loss objective. We report Top-1 and Top-5 (in %) accuracy on ImageNet linear evaluation from 100 epochs pre-trained ResNet-50 backbone on ImageNet 1000 classes.

Method	Top-1 Accuracy	Top-5 Accuracy
Mask Loss		
α_base = 0.3	51.3	77.4
α_base = 0.5	53.9	79.4
α_base = 0.7	54.6	79.8
Hybrid Loss		
λ_base = 0.3	55.0	79.4
λ_base = 0.5	57.8	81.7
λ_base = 0.7	58.2	81.8

**Table 5 sensors-22-05169-t005:** The impact of mask quality on HARL framework performance on the downstream object detection and instance segmentation task. We use Faster R-CNN with C4 backbone for object detection and Mask-RCNN with FPN backbone for instance segmentation. Object detection and instance segmentation backbones are initialized with the 100-epoch pre-trained ResNet-50 backbone on ImageNet dataset. The best result is bolded.

Method	Object Detection						Instance Segmentation		
	VOC07 + 12 Detection			COCO Detection			COCO Segmentation		
	AP_50_	AP	AP_75_	AP_50_	AP	AP_75_	${\mathrm{AP}}_{50}^{mask}$	${\mathrm{AP}}^{mask}$	${\mathrm{AP}}_{75}^{mask}$
HARL (DRFI Masks)	**82.3**	55.4	61.2	44.2	24.6	24.8	41.8	24.3	25.1
HARL (Deep Learning Masks)	82.1	**55.5**	**61.7**	**44.7**	**24.7**	**25.3**	**42.3**	**24.6**	**25.2**

## Data Availability

In this study, we construct two novel sets of heuristic binary mask datasets for the ImageNet ILSVRC training set, which can be found here: https://www.hh-ri.com/2022/05/30/heuristic-attention-representation-learning-for-self-supervised-pretraining/ (accessed on 30 May 2022).

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
