# Peer review of "Heuristic Attention Representation Learning for Self-Supervised Pretraining"

_sensors, 2022, doi:10.3390/s22145169_

Round 1

Reviewer 1 Report

This manuscript proposes a new self-supervised visual representation learning framework by leveraging visual attention with the heuristic binary mask named Heuristic Attention Representation Learning (HARL). This self-supervised framework relies on the joint embedding architecture in which the two neural networks are trained to produce similar embedding for different augmented views of the same image. It adopts prior visual attention object-level by generating a heuristic mask proposal for each training image and maximizes the abstract object-level embedding on vector space instead of whole image representation from previous works. The author has done a lot of experiments to prove that the performance of harl exceeds the existing self-supervised learning methods. The following concerns should be addressed.

1.In the experiment, there seems to be a lack of several advanced self-supervised learning methods, such as DINO and MOCOv3. The reviewers hope that the author can add them to the comparative experiment.

2.In all experiments, the reviewer did not see the flops, params and operation time of the model trained under different self-monitoring methods. I hope the author can add these elements to prove the effectiveness of the model.

3.In all experiments, the backbone chosen by the author is resnet-50. The reviewer suggested that the author add the Vit model as a comparative experiment for the backbone when comparing with the existing advanced methods such as Mocov3 and DINO.

Reviewer 2 Report

This paper considered a self-supervised learning framework (HARL) that maximizes the similarity agreement of object-level latent embedding on vector space across different augmented views of the same image. The paper contains a significant amount of experimental results. The paper is also well organized. The utilization of two heuristic mask proposal techniques from conventional computer vision and unsupervised deep learning methods to generate a binary mask for the natural image dataset is significant.

So I recommend the paper.

Author Response

There is no any questions from reviewer 2.

We would like to thank reviewer 2 to give us positive response and support for this manuscript.

Reviewer 3 Report

This papers designs a network to boost the performance for self-supervised learning task. However, I do have some concerns about this paper, which I have listed as below.

(1). I am wondering what is the robustness of this method to the mask quality. Say if the mask quality is bad due to the inaccurate segmentation mask, will it result in degraded performance. I think an experiment is needed or a paragraph can be added to discuss this.

(2). For Table 2, different Table 1 and 3, why only two methods are included?

(3). I found the performance improvement over BYOL is not significant. Could you elaborate more on the advantage of the proposed method over BYOL?

(4). For tables, I suggest the second best value can be underlined to better deliver the result instead of just bolding the highest value. And the format of Table 2 is messed up. Please make sure the table show on the same page.

(5). "Perceptual grouping" needs an explanation when it first appears in line 70.

(6). The quality of some images are low, such as Figure 3 and 4, please include high resolution images.

(7). Please fix the format in line 195.

(8). stop_gradient is not really a word. Please find a better way instead of using stop_gradient. 

Round 2

Reviewer 3 Report

I appreciate that the authors have taken my suggestions into consideration, and I believe the current version is an improved version.